

# Phonon redshift and Hubble friction in an expanding BEC

Stephen Eckel[1⋆] and Ted Jacobson[2†]

**1** Sensor Sciences Division, National Institute of Standards and Technology,
Gaithersburg, MD 20899, USA
**2** Maryland Center for Fundamental Physics, University of Maryland,
College Park, Maryland 20742, USA

⋆ stephen.eckel@nist.gov, † jacobson@umd.edu

## Abstract

We revisit the theoretical analysis of an expanding ring-shaped Bose-Einstein condensate. Starting from the action and integrating over dimensions orthogonal to the phonon's direction of travel, we derive an effective one-dimensional wave equation for azimuthally-travelling phonons. This wave equation shows that expansion redshifts the phonon frequency at a rate determined by the effective azimuthal sound speed, and damps the amplitude of the phonons at a rate given by $\dot{\mathcal{V}}/\mathcal{V}$, where $\mathcal{V}$ is the volume of the background condensate. This behavior is analogous to the redshifting and "Hubble friction" for quantum fields in the expanding universe and, given the scalings with radius determined by the shape of the ring potential, is consistent with recent experimental and theoretical results. The action-based dimensional reduction methods used here should be applicable in a variety of settings, and are well suited for systematic perturbation expansions.



# 1  Introduction

Sound waves in a fluid resemble light waves in spacetime, and the analog spacetime is curved if the fluid is inhomogeneous in space and/or time. The analogy extends to the level of quantum field theory if the fluid is a coherent quantum system, like a Bose-Einstein condensate (BEC) near zero temperature. This opens the possibility of studying, in a laboratory setting, phenomena closely related to those of quantum field theory in curved spacetime, such as particle creation in an expanding universe and Hawking radiation from a black hole [1], the Unruh effect [2], and false vacuum decay [3].

Unruh's 1981 proposal [4] that Hawking radiation from a black hole could be emulated in a fluid with a sonic horizon has recently been spectacularly realized by Steinhauer, using a quasi-one-dimensional BEC [5–7]. This experimentally establishes the analogy, not only at the classical level of wave propagation, but also at the level of fluctuations of the quantum phonon field in its ground state — i.e. in its "vacuum". Many proposals for simulating cosmological quantum field theory have been put forth as well [3, 8–15]. Experiments have not yet fully realized the analogy, but related phenomena have been observed. For instance, by modulating either the density or interaction strength between the atoms, Sakharov oscillations [16] and the dynamical Casimir effect have been observed [17, 18], and a recent experiment [19] used an expanding BEC ring to simulate several classical field effects in an expanding universe.

In the present paper we focus on phonon propagation on an expanding BEC ring. In the experiment of [19], classical phonons were imprinted onto the condensate, and subsequently the central ring radius $R$ was increased. During such expansion the phonon frequency redshifts, and a "Hubble friction" damps the amplitude of phonon field amplitude at a rate proportional to $\dot{R}/R$. Ref. [19] used an effective one-dimensional wave equation to describe this damping effect, and predicted a constant of proportionality $\gamma_H = 1$, roughly $2\sigma$ higher than the (uncertain) experimental observation of $\gamma_H = 0.55(21)$. More recently, the dynamics of these phonon modes was revisited using conventional Bogoliubov-de Gennes theory [20], and a different prediction, $\gamma_H = 4/7 \approx 0.57$, was found, which is remarkably close to the experimental best fit value.

Inspired by this recent theoretical result of [20], we returned to the dimensional reduction approach that had been taken in [19], aiming to discover if it was in error, and if so to repair it. Our starting point is the action for phonons. By integrating over the dimensions of the action orthogonal to the wavevector of the phonon, now taking into account the changing transverse profile of the condensate, we derive an effective one-dimensional action and wave equation. This predicts $\gamma_H = 4/7$ for the experiment of Ref. [19], in agreement with the recent result of Ref. [20] which uncovered the importance of the changing transverse profile. We justify our dimensional reduction procedure by considering a more complete expansion of the transverse phonon modes, and we analyze possible corrections to the reduced action in the context of perturbation theory. These results provide a clear methodology for deriving effective wave equations for phonons, which can be improved by going to higher orders in perturbation theory, and they clarify the nature of the Hubble friction. In particular, we find

that the Hubble friction in an expanding ring is given by $\dot{\mathcal{V}}/\mathcal{V}$, where $\mathcal{V}$ is the volume of the background condensate, in close parallel with the Hubble friction for a scalar field in cosmology.

This paper is organized as follows. Section 2 reviews the action formalism for both the BEC ground state and the phonon excitations. Section 3 applies this formalism to the case of sound travelling in an elongated, cylindrical condensate and introduces our dimensional reduction technique, first by a simple analysis assuming the modes are constant in the transverse directions, and next by a systematic transverse mode expansion justifying the simple approach. This analysis reproduces the known features of sound in a condensate in a static cylindrical harmonic trap [21–23]. Section 4 applies the same methodology to the case of an expanding ring potential, beginning with an analysis of how the BEC properties scale with the central radius of the ring. Again a quick and efficient dimensional reduction yields the effective one-dimensional azimuthal mode equation, and next a systematic mode analysis justifies it. We illustrate and quantify the accuracy of the approximations using the example of a two-dimensional condensate in an expanding harmonic ring trap. Appendix A proves that the velocity field in a finite, simply connected BEC is uniquely determined by the density, and Appendix B studies the corrections to the scaling relations due to the motion of the condensate.

## 2 Action and field equation for BEC perturbations

The mean field dynamics of a zero temperature BEC are determined by the action for the Gross-Pitaevskii (GP) wave function $\Psi$ [24],

$$\int dt d^3x \sqrt{h}\left[i\hbar\Psi^*\partial_t\Psi - \frac{\hbar^2}{2M}h^{ij}\partial_i\Psi^*\partial_j\Psi - V(x^i,t)\Psi^*\Psi - \frac{g}{2}(\Psi^*\Psi)^2\right]. \qquad (2.1)$$

Here $x^i$ are general time-independent spatial coordinates, $h^{ij}$ is the inverse of the metric tensor $h_{ij}$ defining the (Euclidean) spatial line element $h_{ij}dx^i dx^j$, $h = \det(h_{ij})$, $M$ is the atomic mass, $V$ is an external potential, and $g = 4\pi\hbar^2 a_s/M$ is the interaction coupling constant, where $a_s$ is the $s$-wave scattering length. The GP equation results from Hamilton's principle, i.e. the condition that the action (2.1) be stationary with respect to variations of $\Psi$.

The dynamics of linearized perturbations of a BEC are determined by the action (2.1), expanded to second order around a stationary point, i.e. around a background solution to the GP equation. Adopting the polar ("Madelung") representation of the GP wavefunction,

$$\Psi = \sqrt{n}e^{i\phi}, \qquad (2.2)$$

we expand the density $n$ and phase $\phi$ around the background,[1]

$$n = n_0 + n_1, \qquad \phi = \phi_0 + \phi_1. \qquad (2.3)$$

In the hydrodynamic regime, i.e. for wavelengths long compared to the healing length, $\xi = 1/\sqrt{8\pi a n_0}$, the terms involving spatial derivatives of the density—the "quantum pressure" terms—can be neglected. In this hydrodynamic approximation, the density perturbation $n_1$ appears only algebraically in the action (as $n_1$ and $n_1^2$), and in the linearized approximation its equation of motion then implies that it is determined by $\phi_1$ via

$$n_1 = -(\hbar/g)(\partial_t + v^i\partial_i)\phi_1, \qquad (2.4)$$

---

[1]The variations $n_1$ and $\phi_1$ are related to $\delta\varphi := \delta\Psi/\Psi_0$ as $\delta\varphi = n_1/2n_0 + i\phi_1$, up to terms of quadratic order in the perturbation.

where $v^i = (\hbar/M)\partial^i\phi_0$ is the velocity of the background condensate. Using (2.4) to eliminate $n_1$, one arrives at a quadratic action involving the phase perturbation alone,

$$S = \frac{\hbar^2}{2g}\int dt\,d^3x\,\sqrt{h}\left\{[(\partial_t + v^i\partial_i)\phi_1]^2 - c^2 h^{ij}(\partial_i\phi_1)(\partial_j\phi_1)\right\}, \qquad (2.5)$$

where $c = \sqrt{gn_0/M}$ is the speed of sound in the background condensate, and we have assumed that the coupling $g$ is constant in time and space. If $g$ is not constant, then it should be moved into the integrand of (2.5).

The action (2.5) can be expressed as the action for a massless scalar field in the background of a curved spacetime metric $g_{\mu\nu}$ (see [1] for a review and many references). That geometric representation provides the analogy with field theory in curved spacetime, and in particular the "Hubble friction," which in part motivates the present analysis. In terms of the hydrodynamic metric, and with arbitrary spacetime coordinates $x^\mu$, the action takes the form

$$S = -\frac{\hbar^2}{2g}\int d^4x\,\sqrt{-g}\,g^{\mu\nu}\partial_\mu\phi_1\partial_\nu\phi_1, \qquad (2.6)$$

with spacetime line element $ds^2 = g_{\mu\nu}dx^\mu dx^\nu$ and, in this context, $g = \det(g_{\mu\nu})$. In terms of the laboratory time coordinate $t$, and spatial coordinates $x^i$, the line element implicit in (2.5) is given by

$$ds^2 = c[-c^2 dt^2 + h_{ij}(dx^i - v^i dt)(dx^i - v^j dt)]. \qquad (2.7)$$

In this effective spacetime geometry, the speed of sound $c$ plays the role of the speed of light.[2] Although this curved spacetime representation underpins the motivating analogy, it plays no direct role in our analysis, which is based instead on the action (2.5)

In the case of a confined BEC, the background density $n_0$ vanishes at the boundary of the condensate. In the Thomas-Fermi approximation, the boundary is a sharp edge, so the domain of integration in the action for the perturbation has a spatial boundary. Nevertheless, no boundary term in the action, and no boundary conditions on the perturbation $\phi_1$, are required in order for Hamilton's principle to hold. After variation of $\phi_1$ in the action (2.5), and integration by parts on the derivatives of $\delta\phi_1$, the total derivative terms are

$$\partial_\mu[U^\mu\delta\phi_1(\partial_t + v^j\partial_j)\phi_1] - \partial_i[c^2 h^{ij}\delta\phi_1(\partial_j\phi_1)], \qquad (2.8)$$

where $U^\mu = \delta^\mu_t + v^i\delta^\mu_i$. Upon integration, the second term vanishes since $c^2$ vanishes at the boundary, and the first term vanishes since $U^\mu$ is tangent to the boundary in spacetime.

## 3 Cylindrical BEC

We apply this action formalism first to analyze the simpler case of a static, cylindrical condensate. Doing so, we recover previously known results in a transparent manner, and prepare for the case of the expanding ring, which is our main target here. A static cylindrical BEC has $v^i = 0$ and $c = c(r)$, where $r$ is the cylindrical radius, so the action (2.5) becomes

$$S = \frac{\hbar^2}{2g}\int dt\,dz\,d\theta\,dr\,r\left\{(\partial_t\phi_1)^2 - c^2[(\partial_z\phi_1)^2 + (\partial_r\phi_1)^2 + r^{-2}(\partial_\theta\phi_1)^2]\right\}. \qquad (3.1)$$

The ground state GP wave function in a cylindrical trap potential falls off exponentially outside some (cylindrical) radius. For the present treatment we shall ignore the details of this

---

[2]If $g$ is not constant, then the $1/g$ factor should be included in the definition of the metric. The metric prefactor $c$ then becomes $c/g = n_0/Mc$. In the literature, the prefactor is usually defined as $n_0/c$.

transition region, and use the Thomas-Fermi (TF) approximation for the background density, which vanishes linearly at a sharp value $r_\mu$ of the radius. The edge of the condensate is determined by the equation $V(r_\mu) = \mu$, where $\mu$ is the chemical potential, which is determined, for a given potential $V(r)$, by the number of atoms per unit length of the cylinder. As will be shown below, if the condensate is narrow compared to the mode wavelength in the $z$ direction, the phase $\phi_1$ for the gapless phonon mode—i.e. the mode whose frequency vanishes as the wave vector goes to zero—is approximately constant in the transverse dimensions. This allows one to perform a "dimensional reduction," assuming $\phi_1$ depends on only $t$ and $z$, and integrating out the transverse coordinates $r$ and $\theta$ to obtain an effective 1+1 dimensional action. Since the perturbation $\phi_1$ is only meaningful inside the background condensate radius $r_\mu$, we include only that domain when carrying out the transverse integration. This results in the dimensionally reduced action

$$S = \frac{\pi r_\mu^2 \hbar^2}{2g} \int dt\, dz \left\{ (\partial_t \phi_1)^2 - c_z^2 (\partial_z \phi_1)^2 \right\}, \tag{3.2}$$

where

$$c_z^2 := \frac{\int d\theta\, dr\, r\, c^2}{\int d\theta\, dr\, r} \tag{3.3}$$

is the transverse average of the square of the 3D sound speed. The field equation for these modes is simply the 1D massless wave equation,

$$\partial_t^2 \phi_1 - c_z^2 \partial_z^2 \phi_1 = 0. \tag{3.4}$$

In this approximation, the gapless phonon mode is dispersionless, and propagates at speed $c_z$. Note, however, that although a solution to (3.4) is a stationary point of the reduced action (3.2), it is *not* a stationary point of the original action (3.1), because of the different $r$-dependence of the coefficients of $(\partial_t \phi_1)^2$ and $(\partial_z \phi_1)^2$ in that action. In the next subsection, we shall quantify and correct this failure to satisfy the full field equation.

## 3.1 Transverse mode expansion for a cylindrical BEC

To show that the gapless mode is approximately constant in the transverse dimensions, to evaluate the corrections to that approximation, and to determine the gapped modes, requires a treatment that allows for the dependence of $\phi_1$ on the transverse coordinates. To this end, we exploit the azimuthal and translational symmetries, and expand $\phi_1$ as

$$\phi_1(t, z, \theta, r) = e^{i(kz + m\theta)} \sum_{n=0}^{\infty} \alpha_n(t) f_{nkm}(r), \tag{3.5}$$

where the functions $f_{nkm}$ are a basis for the radial functions, to be chosen shortly as eigenfunctions of a certain differential operator.[3] This is the method used in [21], although there the time dependence was restricted to be harmonic, $e^{-i\omega t}$, which is not an option for the non-stationary expanding ring background.

Inserting the expansion (3.5) into the action (3.1) we obtain, after integration by parts,

$$S = \frac{\hbar^2}{2g} \sum_{n,n'} \int dt\, dz\, d\theta\, dr\, r$$
$$\alpha_n f_{nkm} \left\{ -\partial_t^2 + \left[ r^{-1} \partial_r c^2 r \partial_r - r^{-2} c^2 m^2 - c^2 k^2 \right] \right\} \alpha_{n'} f_{n'km}. \tag{3.6}$$

---

[3]Since $\phi_1$ is by nature a real function, we actually want the real part of the above expression. Alternatively, we can replace one of the $\phi_1$ factors in the quadratic action by $\phi^*$.

For each choice of $k$ and $m$, the operator in the square brackets is Hermitian on the Hilbert space of functions of $r$, with the integration measure $r\,dr$, on the $r$ interval $[0, r_\mu]$, where $c^2(r_\mu) = 0$ defines the edge of the condensate (at which the background density, which is proportional to $c^2$, vanishes). The boundary terms that would have otherwise spoiled the hermiticity property vanish at $r = 0$ and at $r = r_\mu$ because of the factor $rc^2$ between the $\partial_r$ operators. For a given $m$ and $k$, the eigenvalue equation

$$[-r^{-1}\partial_r c^2 r\partial_r + m^2 r^{-2}c^2 + k^2 c^2]f_{nkm} = \lambda_{nkm}f_{nkm}, \tag{3.7}$$

has a discrete set of regular solutions $f_{nkm}$ on the interval $(0, r_\mu)$, with eigenvalues $\lambda_{nkm}$. These functions are orthogonal with respect to the given measure, and can be normalized, so that

$$\int dr\, r\, f_{nkm}f_{n'km} = \delta_{nn'}. \tag{3.8}$$

If we use them as our basis functions the action becomes

$$S = \frac{\pi\hbar^2}{2g}\sum_n \int dt\,dz\, \alpha_n\left\{-\partial_t^2 - \lambda_{nkm}\right\}\alpha_n. \tag{3.9}$$

Stationarity of the action with respect to $\alpha_n$ variations then implies that the mode labeled by $(k, m, n)$ is a harmonic oscillator, with frequency

$$\omega_n^2 = \lambda_{nkm}. \tag{3.10}$$

This is gives the complete spectrum of phonon modes, within the linearized hydrodynamic approximation, and neglecting the edge effects that go beyond the TF approximation. If, instead of the TF approximation, we take for the background the *exact* solution of the GP equation, then $c^2$ falls exponentially to zero, so does not strictly vanish at any finite radius. Presumably instead of considering the eigenvalue problem for (3.7) on the finite TF domain, it can be considered on the half-line $[0, \infty)$, with the boundary condition that the mode functions vanish as $r \to \infty$.

For each $(k, m)$, there is a tower of transverse modes, given by the eigenfunctions $f_{nkm}$ defined in (3.7). We restrict attention here to azimuthally symmetric modes, $m = 0$, and suppress the $m = 0$ label. For $k = 0$ the lowest mode has constant eigenfunction, $f_0 = $ const, and eigenvalue $\lambda_0 = 0$. (For the $k = 0$ modes we suppress also the $k = 0$ label.) This constant mode corresponds to an overall global phase shift of the condensate wavefunction, which has no physical meaning. The phonon is the "Goldstone boson" corresponding to this global shift symmetry, which is spontaneously broken by the condensate wave function. For nonzero $k$, the lowest eigenfunction $f_{0k}$ corresponds to a 'gapless' phonon mode. To find its frequency $\omega_{0k}$, we must solve the $k$-dependent eigenvalue problem (3.7). For $kr_\mu \ll 1$, the $c^2 k^2$ term can be treated as a perturbation. According to the standard method of quantum mechanical perturbation theory for eigenvalues, the first order perturbation of the eigenvalue is

$$\lambda_{0k}^{(1)} = k^2\langle f_0|c^2|f_0\rangle, \tag{3.11}$$

where $f_0 \propto 1$ is the unperturbed eigenstate. To order $k^2$, this yields the linear dispersion relation we obtained via dimensional reduction of the action, with sound speed (3.3) given by the transverse average of the local speed $c$.

The second order correction to the eigenvalue is

$$\lambda_{0k}^{(2)} = -k^4\sum_{n>0}|\langle f_n|c^2|f_0\rangle|^2/\lambda_n. \tag{3.12}$$

This shows that the mode is dispersive, of order $k^2 r_\mu^2$ relative to the leading, nondispersive term. The negative sign is familiar from quantum mechanical perturbation theory: the second order perturbation of the ground state energy is *always* negative (or zero). The corresponding effect on the transverse wave profile is to deform the mode function $f_{0k}$ so that it is smaller at the center and larger toward the edges of the condensate [23]. This can be understood qualitatively using the variational principle: the exact eigenvalue $\lambda_{0k}$ is the minimum of the expectation value of the "Hamiltonian," $\int dr\, r\, c^2[(\partial_r f)^2 + k^2 f^2]$, with $f$ normalized as in (3.8). For $k = 0$, this is minimized by a constant $f$. As $k$ grows, the integral can be decreased by concentrating $f$ more in the outer region where $rc^2$ is smaller. If $k$ is large enough for the quantum potential to be nonnegligible, this hydrodynamic description breaks down, and the dispersion eventually approaches the free particle one [23].

The lowest transverse mode function $f_{0k}$ is constant only in the $k \to 0$ limit. The first order correction brings in transverse dependence, which can be calculated, according to standard quantum mechanical perturbation theory, as

$$f_{0k}^{(1)} = k^2 \sum_{n>0} \frac{\langle f_n | c^2 | f_0 \rangle}{-\lambda_n} f_n. \tag{3.13}$$

The higher transverse modes of the $k = 0$ mode are thus mixed in with the constant piece, for any nonzero $k$.

## 3.2 Harmonic radial trap

Let us illustrate the preceding results for the case of a harmonic radial confining potential, $V(r) = \frac{1}{2} M \omega_r^2 r^2$. The TF sound speed in the cylindrical condensate with confining potential $V$ and chemical potential $\mu$ is $c^2 = (\mu - V)/M = \frac{1}{2} \omega_r^2 (r_\mu^2 - r^2)$. Adopting units with $\omega_r = r_\mu = 1$, the $k = m = 0$ eigenfunctions defined in (3.7) become (up to normalization) the Legendre polynomials $P_n(x)$, with $x = 2r^2 - 1$ [21]. In particular, the first two normalized eigenfunctions are

$$f_0 = \sqrt{2}, \qquad f_1 = \sqrt{6}\left(1 - 2r^2\right), \tag{3.14}$$

with corresponding eigenvalues

$$\lambda_0 = 0, \qquad \lambda_1 = 4. \tag{3.15}$$

The effective axial sound speed defined by $c_z^2 = \langle f_0 | c^2 | f_0 \rangle$ (3.3,3.11) is then given by

$$c_z = 1/2, \tag{3.16}$$

which is $1/\sqrt{2}$ times the local sound speed, $c(0) = 1/\sqrt{2}$, at the center of the condensate. The second order correction $\lambda_{0k}^{(2)}$ to the squared frequency can now be easily evaluated analytically using (3.12). Note that $c^2$ in this condensate can be written as a superposition of the first two eigenfunctions, $c^2 = af_0 + bf_1$, with coefficients $a = 1/\sqrt{32}$ and $b = 1/\sqrt{96}$. The orthogonality of the $f_n$ implies that only the $f_1$ term contributes to the sum (3.12), so the sum is just $b^2 f_0^2 / \lambda_1 = 1/192$. The correction to the eigenvalue is thus

$$\lambda_0^{(2)} = -k^4/192 = -(kc_z)^2(kr_\mu)^2/48,$$

in agreement with the result reported in Ref. [22]. Similarly, from (3.13) we obtain the first correction to the constant transverse profile of the lowest mode,

$$f_{0k}^{(1)} = -\frac{(kr_\mu)^2}{16\sqrt{3}} f_1. \tag{3.17}$$

These results justify the dimensional reduction of the action that was employed, at the beginning of this section, to derive the effective one-dimensional wave equation for the amplitude $\alpha$. We assumed there that the mode was strictly constant in the transverse directions. Here we see that, for the example of a cylindrical harmonic trap with a condensate of radius $r_\mu$, a mode with axial wave vector $k$ is nearly constant in the transverse directions if $kr_\mu \ll 1$. In particular, we have $\partial_z \phi_1 \sim k f_0 \sim k$ and, from (3.17) and (3.14), $\partial_r \phi_1 \sim k^2 \partial_r f_1 \sim k^2 r$, and the ratio of the latter to the former is $\sim kr \le kr_\mu$. The transverse derivative term is thus suppressed by $(kr_\mu)^2$ in the action (3.1) compared to the axial derivative term. Note, however, that in the *three-dimensional* equation of motion the transverse term is of *equal importance*, because $\partial_z^2 \phi_1 \sim k^2 f_0$ and $\partial_r^2 \phi_1 \sim k^2 \partial_r^2 f_1$, both of which are $O(k^2)$. This equal importance is to be expected, since $c(r)$ has O(1) dependence on $r$ within the cylinder, so the only way that $\phi_1$ can satisfy the three-dimensional field equation is if the transverse derivative terms take this variation of $c(r)$ into account. Although discussed here for the particular case of a harmonic trap, these scaling properties would hold for a wide class of trapping potentials.

## 4 Expanding Ring BEC

In this section we analyze the modes on an expanding ring BEC. First we determine the evolution of the background on which the perturbations propagate. Next we give a brief derivation of the Hubble friction effect, using dimensional reduction of the action. Finally, we examine the mode decomposition in order to further justify the dimensional reduction procedure.

### 4.1 Scaling behavior of the expanding ring condensate

The confining potential for the expanding ring has the form[4]

$$V(r,z,t) := V(r - R(t), z) = V(\rho, z), \tag{4.1}$$

where $\rho$ is a comoving cylindrical radius coordinate,

$$\rho := r - R(t). \tag{4.2}$$

As a first approximation to the expanding condensate, we find the instantaneous ground state in the Thomas-Fermi (TF) approximation. In this approximation, the velocity of the condensate as well as the quantum pressure are neglected, which amounts to neglecting the spatial derivative terms in (2.1). The static approximation is clearly justified if $\dot{R} \ll c$ and $v \ll c$, where $v$ is the local "contraction" velocity of the condensate relative to the center of the expanding ring potential. These conditions are not strictly required; in fact, the first of these conditions does not hold for the condensate in the experiment of Ref. [19]. The static approximation is nevertheless surprisingly accurate, and allows us to derive simple formulas for how the condensate properties scale with $R$ as the ring expands. In Appendix B, we consider two non-static corrections for these scalings. In short, inertial forces perturb the shape and position of the background condensate through the additional inertial potential, which we estimate by considering two limiting cases. For the case of the a harmonic potential, the leading order effect is captured by scaling with respect to the shifted radial position of the condensate, rather than the center of the potential. For the case of the square well potential, the effect is suppressed by the thin-ring parameter, defined below. We estimate the non-static error for the experiment of Ref. [19] at <15 %. Even faster expansions excite the first order radial Bogoliubov mode, but

---

[4]In the experiment of [19], the potential is produced by optical fields controlled with a digital micromirror device. Here, we neglect the azimuthal imperfections that were present in the experimental potential.

because this mode function is odd about the center of the condensate, its effect is zero when the excitation amplitude is small. For large excitation amplitudes, such as those evidenced by the large amplitude center-of-mass oscillation at the end of the $R_f/R_i = 1.9$ expansion of Ref. [19], the effect would be suppressed by the thin ring parameter and averaging over a full oscillation.

After neglecting the spatial derivatives, and assuming stationary time dependence, $i\hbar\partial_t\Psi = \mu\Psi$, the GP equation implies the relation

$$\mu = gn + V(\rho, z), \tag{4.3}$$

between the chemical potential $\mu$, the number density $n$, and the trap potential $V$.[5] Solving for the total number of atoms $N$ using Eq. (4.3) yields

$$gN = 2\pi \int d\rho\, dz\, (R + \rho)[\mu - V(\rho, z)], \tag{4.4}$$

where the integral is over the region in which the condensate density is nonzero, i.e. where $V(\rho, z) < \mu$. Equation (4.4) determines the value of $\mu$, for a given number of atoms $N$ and potential radius $R$. As the ring potential expands, $\mu$ and $n$ change with the radius $R$ of the potential, but $N$ is fixed.

We now assume that the potential is a sum of powers of $\rho$ and $z$,

$$V(\rho, z) = \mu[(\rho/\rho_\mu)^{n_\rho} + (z/z_\mu)^{n_z}], \tag{4.5}$$

with $n_\rho$ and $n_z$ even, positive, integers. We have expressed $V$ in terms of the TF radii $\rho_\mu$ and $z_\mu$ that demarcate the edge of the condensate if the chemical potential is $\mu$. Since $V$ is independent of $\mu$, the $\mu$ dependence in (4.5) must cancel, so evidently the TF radii scale with $\mu$ as

$$\rho_\mu \propto \mu^{1/n_\rho} \quad \text{and} \quad z_\mu \propto \mu^{1/n_z}. \tag{4.6}$$

In terms of dimensionless integration variables

$$\bar{\rho} \equiv \rho/\rho_\mu \quad \text{and} \quad \bar{z} \equiv z/z_\mu, \tag{4.7}$$

(4.4) with $V$ given by (4.5) becomes

$$gN = 2\pi R \rho_\mu z_\mu \mu \int d\bar{\rho}\, d\bar{z}(1 + \zeta\bar{\rho})(1 - \bar{\rho}^{n_\rho} - \bar{z}^{n_z}), \tag{4.8}$$

where $\zeta = \rho_\mu/R$ is the ring thickness parameter.[6] The integral in (4.8) is over the region where the integrand is positive, and the contribution from the $\zeta\rho$ term vanishes by the symmetry of $V$ under $\bar{\rho} \to -\bar{\rho}$. The integral is therefore independent of $R$ and $N$, so is a pure number determined by $n_\rho$ and $n_z$, whose value we do not need here.[7] It thus follows, for fixed $N$, that

$$\mu \propto (R\rho_\mu z_\mu)^{-1}, \tag{4.9}$$

which together with (4.6) implies the scaling of the chemical potential,

$$\mu \propto R^{-\alpha}, \quad \text{with} \quad \alpha = \frac{1}{1 + 1/n_\rho + 1/n_z}. \tag{4.10}$$

---

[5]Note that a constant shift of $V$ produces the same constant shift of $\mu$, with no effect on $n$.

[6]For the experiment of Ref. [19], $\zeta \approx 0.4 \to 0.2$ for the $R_f/R_i = 1.9(1)$ expansion and $\zeta \approx 0.6 \to 0.1$ for the $R_f/R_i = 4.1(3)$ expansion.

[7]The value is $4\Gamma(1 + \frac{1}{n_\rho})\Gamma(1 + \frac{1}{n_z})/\Gamma(2 + \frac{1}{n_\rho} + \frac{1}{n_z})$. For the experiment of Ref. [19], where $n_\rho = 4$ and $n_z = 2$, this is $\approx 1.998$.

Given the scaling of $\mu$ with $R$, that of other quantities can be determined. The density is given by

$$gn = \mu(1 - \bar\rho^{n_\rho} - \bar z^{n_z}),\tag{4.11}$$

so that the local sound speed is given by

$$c^2 = gn/M \propto \mu \propto R^{-\alpha}.\tag{4.12}$$

The volume of the condensate scales as

$$\mathcal{V} \propto R\rho_\mu z_\mu \propto R^\alpha,\tag{4.13}$$

and the cross-sectional area scales as

$$\mathcal{A} \propto \rho_\mu z_\mu \propto R^{-\beta}, \quad \text{with} \quad \beta = \alpha(1/n_\rho + 1/n_z).\tag{4.14}$$

The condensate contracts in $\rho$ and $z$ as the ring expands, because the decrease of density lowers the effect of the interatomic repulsion.

For the experiment discussed in [19], the average radial potential was found to have the form $V(\rho) = M\omega_\rho^2\rho^2/2 + \lambda\rho^4$, with coefficients of $\omega_r \approx 2\pi \times 100$ Hz and $\lambda/\hbar \approx 2\pi \times 0.8$ Hz$/\mu$m$^{-4}$. The corresponding Thomas-Fermi radius for this combined quartic plus quadratic potential, at $z = 0$ and to lowest order in $M\omega_p^2/\sqrt{\mu\lambda}$, is

$$\rho_\mu \approx \left(\frac{\mu}{\lambda}\right)^{1/4}\left[1 - \frac{1}{8}\frac{M\omega_\rho^2}{\sqrt{\mu\lambda}}\right].\tag{4.15}$$

In the experiment of [19], the relative magnitude of the change in the TF radius due to the presence of the quadratic term in the potential is less than 10 % for all values of $R$. Thus, for the purposes of determining the scaling, we neglect the quadratic term. With $n_\rho = 4$ and $n_z = 2$, , (4.10) yields $\alpha = 4/7$. Were the quadratic correction included, we would expect the effective $\alpha$ to be slightly less than the predicted $4/7$, but no smaller than the expected $\alpha = 1/2$ for a quadratic potential. For the two-dimensional simulations reported in [19], the scaling is obtained by sending $n_z \to \infty$, which results in $\alpha = 4/5$.

## 4.2 Hubble friction from the azimuthal reduced action

We now consider azimuthal perturbations of an expanding BEC ring. By dimensional reduction of the action from 3D to an effective azimuthal action, we quickly obtain the effective azimuthal mode equation, including the Hubble friction coefficient.

In cylindrical coordinates $(r, \theta, z)$, the action (2.5) for a general phase perturbation of the expanding ring condensate takes the form

$$S = \frac{\hbar^2}{2g}\int dt\,d\theta\,dz\,dr\,r\left\{[(\partial_t + (\dot R + v^\rho)\partial_r + v^z\partial_z)\phi_1]^2 - c^2h^{ij}(\partial_i\phi_1)(\partial_j\phi_1)\right\},\tag{4.16}$$

where $R(t)$ is the central radius of the ring potential (4.1), and $v^\rho$ and $v^z$ are the local velocity components of the condensate relative to the center of the ring. In terms of the co-moving radial coordinate, $\rho = r - R(t)$, the action becomes

$$S = \frac{\hbar^2}{2g}\int dt\,d\theta\,dz\,d\rho\,(R(t) + \rho)\left\{[(\partial_t + v^\rho\partial_\rho + v^z\partial_z)\phi_1]^2 - c^2h^{ij}(\partial_i\phi_1)(\partial_j\phi_1)\right\}.\tag{4.17}$$

In analogy to the case of the cylinder condensate, the phase perturbation for azimuthal phonons on the gapless branch can be taken to depend, in the first approximation, on only $t$ and $\theta$. (In

the following subsection this assumption will be justified more carefully.) Integrating over the transverse dimensions out to the edge of the condensate then yields the dimensonally reduced action,

$$S = \frac{\hbar^2}{4\pi g} \int dt d\theta \, \mathcal{V} \left\{ (\partial_t \phi_1)^2 - c_\theta^2 R^{-2} (\partial_\theta \phi_1)^2 \right\}, \tag{4.18}$$

where

$$\mathcal{V} := 2\pi \int dz d\rho \, (R + \rho), \quad \text{and} \quad c_\theta^2 := \frac{2\pi R}{\mathcal{V}} \int dz d\rho \, c^2 / (1 + \rho/R), \tag{4.19}$$

are the (time-dependent) volume of the condensate and the 'average' of the sound speed. In the thin ring limit, $\rho/R$ can be neglected, so $c_\theta^2$ becomes precisely the transverse-averaged sound speed, as with the cylindrical condensate. In this case, the integrals can be evaluated in closed form in terms of $\Gamma$ functions, and for all even powers $n_\rho$ and $n_z$ in the potential (4.5) we find

$$c_\theta^2 = \alpha c_0^2 \qquad \text{(thin ring)}, \tag{4.20}$$

where $\alpha$ is the combination of $n_\rho$ and $n_z$ defined in (4.10), and $c_0 = \sqrt{\mu/M}$ is the speed of sound in the center of the condensate. For a condensate that is symmetric under $\rho \to -\rho$, the $\rho$ term does not contribute to $\mathcal{V}$, nor does it contribute to $c_\theta^2$ when the denominator is expanded as $(1 + \rho/R)^{-1} = 1 - \rho/R + \rho^2/R^2 + \cdots$. In this case, the correction to (4.20) is of order $\zeta^2$.

Note that (4.18) *cannot* be expressed as a scalar field minimally coupled to some two-dimensional spacetime metric, i.e. as $\int d^2x \sqrt{-g} g^{\alpha\beta} \partial_\alpha \phi_1 \partial_\beta \phi_1$. This is because, for *any* metric $g_{\alpha\beta}$, the tensor density $\sqrt{-g} g^{\alpha\beta}$ has determinant $-1$. In the action (4.18), this determinant corresponds to the product of the coefficients of $(\partial_t \phi_1)^2$ and $(\partial_\theta \phi_1)^2$, which is proportional to the time-dependent quantity $-(\mathcal{V} c_\theta)^2$.

According to the reduced action (4.18), the field equation for the azimuthal phonon mode on the expanding ring is

$$\partial_t^2 \phi_1 + \frac{\dot{\mathcal{V}}}{\mathcal{V}} \partial_t \phi_1 - c_\theta^2 R^{-2} \partial_\theta^2 \phi_1 = 0. \tag{4.21}$$

As shown in the previous subsection, for a power law potential in the adiabatic TF approximation we have $\mathcal{V} \propto R^\alpha$, where $\alpha$ is determined by the power law exponents, (4.10). In this case, $\dot{\mathcal{V}}/\mathcal{V} = \alpha \dot{R}/R$, and (4.21) becomes

$$\partial_t^2 \phi_1 + \alpha \frac{\dot{R}}{R} \partial_t \phi_1 - c_\theta^2 R^{-2} \partial_\theta^2 \phi_1 = 0. \tag{4.22}$$

The $\partial_t \phi_1$ term in (4.21) is identical to the "Hubble friction" term in the field equation for a minimally coupled scalar field in an expanding homogeneous universe. In the isotropic case, with three spatial dimensions, that term would have coefficient $3\dot{a}/a$, where $a(t)$ is the scale factor of the spatial sections of the universe.

For solutions of the form $\phi_1 \propto e^{im\theta}$, the last term in (4.22) becomes $m^2 c_\theta^2 R^{-2} \phi_1$, which identifies the stationary mode frequency $\omega_m = m c_\theta / R$. As the ring expands, the modes redshift not only because $R$ grows but also because the density, and therefore $c_\theta$, drops. In the thin ring limit, (4.20) yields $c_\theta \propto \mu^{1/2}$, so using (4.10) we find the frequency scaling $\omega_m \propto R^{-\alpha/2-1}$. For the experiment of [19] this yields $\omega_m \propto R^{-9/7}$, and (4.20) yields $c_\theta = \sqrt{4/7} c_0$. These results are in agreement with those of Ref. [19].

The full effect of the changing condensate volume $\mathcal{V}(t)$ (4.5) was missed in Ref. [19], which accounted only for the increase of $R(t)$ and neglected the decrease in transverse area $\mathcal{A}(t)$ (4.14). Ref. [19] therefore predicted $\gamma_H = 1$ for the Hubble friction coefficient $\gamma_H \dot{R}/R$, which failed to match the experimentally measured (but uncertain) best fit value of $\gamma_H = 0.55$. It was subsequently shown in Ref. [20] that the deformation of the condensate modifies the evolution

of the azimuthal mode, in precisely the way we have rederived here using a different method. For the potential in the experiment of Ref. [19], the predicted coefficient is $\gamma_H \approx 4/7 \approx 0.57$, which is remarkably close to the best fit measured value. For the two-dimensional simulation of Ref. [19], the exponents in the potential were $n_\rho = 4$ and, effectively, $n_z = \infty$, which yields $\gamma_H = 4/5$. No best fit value was extracted from the simulation, although the (incorrect) prediction $\gamma_H = 1$ compared rather well with the simulation. Looking more closely, however, the discrepancy can be discerned.

It is worth clarifying the relation between the phase perturbation $\phi_1$ and the experimental observable. What can be directly and accurately measured is the angular density perturbation $n_\theta$, which is the 3D density perturbation $n_1$ integrated over $dz$ and $d\rho$, weighted by $R(t) + \rho$.[8] For an azimuthal mode, (2.4) implies $n_1 = -(\hbar/g)\partial_t \phi_1$, since the velocity $v^i$ has no $\theta$ component. It follows that $n_\theta \propto \mathcal{V}\partial_t\langle\phi_1\rangle$, where $\langle\phi_1\rangle$ is the transverse average of $\phi_1$. For the analysis of mode amplitude damping in both the experiment and simulations of [19], $n_\theta$ was divided by $\mathcal{V}$, which was found using the Thomas-Fermi model described in the previous subsection. This yielded the transversely averaged 3D density perturbation, $\langle n_1\rangle$, which according to (2.4) is $\propto \langle\partial_t\phi_1\rangle$. Thus, although the density and phase differ by a time derivative, the cumulative effect of the Hubble friction during the expansion, inferred for $\langle n_1\rangle$, should agree with that found for $\langle\phi_1\rangle$. In the approximation of transverse uniformity for the lowest mode, the latter is governed by the effective 1D mode equation, (4.21).

## 4.3 Transverse mode expansion for an expanding ring condensate

We turn now to a systematic mode expansion of the linear perturbations of an expanding ring BEC. In the previous subsection we simplified the problem by assuming, motivated by the behavior in the static cylinder case, that the lowest mode is constant in the transverse directions, in the radially comoving coordinate system. This assumption eliminates the spatial derivative terms that are multiplied by the flow velocity. The primary aim of this section is to assess the accuracy of that simplifying assumption.

The principal complication comes from the velocity dependent terms in the convective derivatives in the action. As shown in Appendix A, since the background flow velocity is irrotational, it is is uniquely determined, apart from possible circulation around noncontractible loops, by the time dependence of the density together with the atom number conservation equation. The density is given in the instantaneous TF approximation by (4.11). In the "comoving" coordinate system $(\bar\rho, \bar z, \theta)$ (4.7), therefore, all of the time dependence of $n$ is in the overall factor $\mu$. In this coordinate system the spatial volume density is

$$\sqrt{h} = \rho_\mu z_\mu (R + \rho_\mu \bar\rho), \tag{4.23}$$

hence the volume weighted number density is

$$n\sqrt{h} = \mu \rho_\mu z_\mu (R + \rho_\mu \bar\rho)[1 - \bar\rho^{n_\rho} - \bar z^{n_z}]/g, \tag{4.24}$$

and the fixed $N$ scaling relation (4.9) then implies

$$n\sqrt{h} \propto 1 + \rho_\mu \bar\rho/R. \tag{4.25}$$

This is independent of time in the thin ring limit, so it follows from the argument in Appendix A that in that limit we have $v^{\bar\rho} = v^{\bar z} = 0$. That is, in the thin ring limit—but only in that limit—the condensate is strictly static in the comoving coordinate system.

---

[8]Specifically, $n_1$ is the measured density difference between a condensate with an imprinted phonon and without, corrected for atom number variations.

For the remainder of this section and the next, we restrict attention to the thin ring limit, and we use the notation

$$\mathcal{W} := \rho_\mu z_\mu R \approx \sqrt{h} \tag{4.26}$$

for the spatial volume density. In this approximation the action (4.17) in comoving coordinates takes the form

$$S = \frac{\hbar^2}{2g} \int dt d\theta d\bar{\rho} d\bar{z} \, \mathcal{W} \Big\{ (\partial_t \phi_1)^2$$
$$- c^2 \Big[ \rho_\mu^{-2}(\partial_{\bar{\rho}} \phi_1)^2 + z_\mu^{-2}(\partial_{\bar{z}} \phi_1)^2 + R^{-2}(\partial_\theta \phi_1)^2 \Big] \Big\}. \tag{4.27}$$

Note that this action has only one symmetry, the azimuthal one, in contrast to the case of the static cylinder which has three symmetries: two spatial and one temporal.

To proceed with a mode analysis, we fix an azimuthal mode number $m$ and expand $\phi_1$ in a basis of transverse functions, as

$$\phi_1(t, \theta, \bar{\rho}, \bar{z}) = e^{im\theta} \sum_n \alpha_n(t) q_n(\bar{\rho}, \bar{z}; t). \tag{4.28}$$

(For notational simplicity the index $m$ that should label the functions $q_n$ is omitted here.) Inserting this expansion into the action (4.27) and integrating by parts yields for the spatial derivative terms

$$\sum_{n,n'} \int dt d\bar{\rho} d\bar{z} \, \mathcal{W} \, \alpha_n q_n [\rho_\mu^{-2} \partial_{\bar{\rho}} c^2 \partial_{\bar{\rho}} + z_\mu^{-2} \partial_{\bar{z}} c^2 \partial_{\bar{z}} - m^2 R^{-2} c^2] \alpha_{n'} q_{n'}. \tag{4.29}$$

This is quite similar to what we had for the stationary cylindrical condensate, (3.6), but here the differential operator in the square brackets depends on time. It is nevertheless Hermitian on the Hilbert space of functions on the support of the background condensate, with respect to the measure $d\bar{z} d\bar{\rho}$, and thus has time-dependent eigenfunctions $q_n(\rho, z; t)$, with real eigenvalues $\lambda_n(t)$,

$$[-\rho_\mu^{-2} \partial_{\bar{\rho}} c^2 \partial_{\bar{\rho}} - z_\mu^{-2} \partial_{\bar{z}} c^2 \partial_{\bar{z}} + m^2 R^{-2} c^2] q_n = \lambda_n q_n. \tag{4.30}$$

The functions $q_n$ are orthogonal, and we choose them to be normalized, viz.

$$\int q_n q_{n'} = \delta_{nn'}, \tag{4.31}$$

where the integration measure $d\bar{\rho} d\bar{z}$ is implicit. The equation of motion following from the action (4.27) thus takes the form

$$\left( \partial_t^2 + \frac{\dot{\mathcal{W}}}{\mathcal{W}} \partial_t + \lambda_n \right) \alpha_n + \sum_{n'} \left( \mathcal{W}^{-1} \{ \partial_t, \mathcal{W} A_{nn'} \} - B_{nn'} \right) \alpha_{n'} = 0, \tag{4.32}$$

where $\{,\}$ is the anticommutator, and

$$A_{nn'} := \int q_n \partial_t q_{n'}, \qquad B_{nn'} := \int \partial_t q_n \, \partial_t q_{n'}. \tag{4.33}$$

Note that $A_{nn'} = -A_{n'n}$ and $B_{nn'} = B_{n'n}$.

Each of the terms in the operator in (4.30) simply scales with a power of $R$ as the ring expands, but the powers differ, so the operator is non-trivially time-dependent. This operator can be further simplified by considering the long wavelength limit, in which the third, $m^2 R^{-2} c^2$, term is treated as a perturbation. Neglecting that term for the moment, the two remaining

terms scale as $R^{-\alpha(1+1/n_\rho)}$ and $R^{-\alpha(1+1/n_z)}$. For the experiment of [19], these exponents are 5/7 and 6/7, respectively. Since these are fairly close, the time dependence of the operator is approximately just an overall scaling, which affects the eigenvalue but not the eigenfunctions. Were these exponents identical, the eigenfunctions would be time-independent, so that $A_{nn'}$ and $B_{nn'}$ would simply vanish when the perturbation $m^2R^{-2}c^2$ is neglected. The lowest mode is the Goldstone mode, which for $m = 0$ is constant in space and has eigenvalue $\lambda = 0$ for all time. The first order perturbation from the $m$ term yields $\lambda_0^{(1)} = m^2R^{-2}c_\theta^2$, where $c_\theta^2 = \langle q_0|c^2|q_0\rangle$ is just the transverse average of the sound speed. Finally, we note that $\dot{\mathcal{W}}/\mathcal{W} = \dot{\mathcal{V}}/\mathcal{V}$ in the thin ring limit. In the approximation just described, the equation of motion (4.32) for the lowest mode amplitude $\alpha_0$ thus agrees with the equation of motion (4.21) derived from the dimensionally reduced action. Beyond the thin ring approximation there will be various corrections whose magnitude could be estimated by a perturbative expansion in the ring thickness parameter. Such a study would be useful, but it lies beyond the scope of the present article.

## 4.4 Expanding two-dimensional harmonic thin ring potential

We now consider as an example a two-dimensional ring condensate with harmonic confinement, again restricting to the thin ring limit. For this example, $V = M\omega^2\rho^2/2 = \mu(\rho/\rho_\mu)^2 =: \mu\bar{\rho}^2$, so the results of Sec. 4.1 imply $\mu \propto R^{-2/3}$, $\rho_\mu \propto R^{-1/3}$, $\mathcal{V} \propto R^{2/3}$, $c^2 = c_0^2(1-\bar{\rho}^2)$, and $c_0^2/\rho_\mu^2 = \omega^2/2$. For vanishing angular mode number $m$, the operator in (4.30) in this example (with no $z$-term) is thus time independent, and the eigenmode solutions are Legendre polynomials, as for the cylindrical condensate treated in Sec. 3.2, but here they are supported on the domain $\bar{\rho} \in (-1,1)$ rather than $\bar{\rho} \in (0,1)$. The first three normalized eigenfunctions are

$$q_0 = \sqrt{\frac{1}{2}}, \qquad q_1 = \sqrt{\frac{3}{2}}\,\bar{\rho}, \qquad q_2 = \sqrt{\frac{5}{8}}(3\bar{\rho}^2-1), \qquad (4.34)$$

with eigenvalues $\lambda_n = n(n+1)\omega^2/2$. Using standard perturbation theory for the $m^2R^{-2}c^2$ term in (4.30),[9] we find the eigenvalue for the lowest mode to second order,

$$\lambda_{0m} = \frac{2}{3}\frac{m^2c_0^2}{R^2} - \frac{2m^4c_0^4}{135R^4\omega^2} = c_\theta^2 k^2\left(1 - \frac{(k\rho_\mu)^2}{45}\right), \qquad (4.35)$$

where $k = m/R$ is the azimuthal wavevector of the excitation and $c_\theta^2 = 2c_0^2/3$, and the corresponding perturbed mode function to first order,

$$q_{0m} = q_0 + \frac{(k\rho_\mu)^2}{9\sqrt{5}}q_2. \qquad (4.36)$$

Because the ring has been assumed to be thin, the first order correction is small unless $m$ is large. This transverse mode is therefore nearly constant for small $m$, justifying the dimensional reduction. Note, however, that the perturbed modes are time dependent, since $\rho_\mu/R \propto R^{-4/3}$, so that $A_{nn'}$ and $B_{nn'}$ (4.33) acquire non-zero values for $m \neq 0$. Since $B_{nn'}$ is quadratic in time derivatives of the modes, it is second order in the perturbation, so to first order only the $A_{nn'}$ term contributes.

We focus here on the lowest mode, whose amplitude $\alpha_0$ is mixed with the other $\alpha_n$ via the $A_{0n}$ term in (4.32). In the experiment of [19], the higher modes are initially only thermally populated, so they have much smaller amplitudes than the imprinted amplitude $\alpha_0$. In that setting, the $A_{0n}$ term can only affect the evolution of $\alpha_0$ to the extent that the other amplitudes $\alpha_n$ have been sourced by $\alpha_0$ via their $A_{n0}$ terms during the evolution. Their effect on the

---

[9]It is convenient to use $c^2 = (\sqrt{2}/3)(q_0 - q_2/\sqrt{5})$, together with the orthonormality relation (3.8).

evolution of $\alpha_0$ will thus be of second order. The lowest order contribution of this sort involves only the product $A_{02}A_{20} = -A_{20}^2$. At the same order there is the contribution of the $B_{00}$ term to the azimuthal sound speed. Using the perturbed eigenfunction (4.36), to lowest order in $k\rho_\mu$ we find that $B_{00} = A_{20}^2$, and

$$A_{20} = \frac{2}{9\sqrt{5}}\partial_t(k\rho_\mu)^2 = -\frac{4(k\rho_\mu)^2}{27\sqrt{5}}\frac{\dot{R}}{R}. \tag{4.37}$$

To assess the relative effect of these terms in (4.32), we should compare $A_{20}$ to $\dot{\mathcal{W}}/\mathcal{W} = 2\dot{R}/3R$ and $B_{00}$ to $\lambda_{0m} = (kc_\theta)^2$. The ratio of the former quantities is $A_{20} : (2\dot{R}/3R) = (2/9\sqrt{5})(k\rho_\mu)^2 \approx 0.1(k\rho_\mu)^2$; hence, even for sufficiently large $m$ where $k\rho_\mu \sim 1$, the effect of $A_{20}$ is suppressed by $\sim (0.1)^2$, i.e. it is at most a 1 % effect. The latter ratio is $B_{00} : (kc_\theta)^2 = (16/1215)(k\rho_\mu)^2$ $(\dot{R}/R)/\omega \approx 0.01(k\rho_\mu)^2(\dot{R}/R)/\omega$. Using the parameters for the $a = 1.9(1)$ expansion found in Ref. [19], we have $(\dot{R}/R)/\omega \approx 0.2$, so as long as $k\rho_\mu < 1$ the effect of $B_{00}$ is at most of order $10^{-3}$. Thus, as anticipated more generally in the previous section, in this example of a thin two-dimensional harmonic ring the mode-mixing due to $A_{nn'}$ and $B_{nn'}$ is indeed a very small effect, as long as the mode number $m$ is not much larger than the reciprocal of the ring thickness parameter.

# 5 Summary and Discussion

We have presented an efficient technique for deriving an effective one-dimensional wave equation for phonons in a Bose Einstein condensate with transverse cross section much smaller than the phonon wavelength, but still much larger than the healing length. The method begins with the action for the phonon field, and proceeds by integrating out the transverse dimensions. At leading order it is especially simple, because of the constant transverse profile of the phonon modes. Carrying out a systematic expansion in the transverse modes, we justified this simple leading order result, and derived corrections perturbatively in the transverse size parameter.

We first applied this method to a condensate in a general static, cylindrical trapping potential. For a harmonic potential we efficiently reproduced known results [21–23], including the effective longitudinal sound speed and higher order finite-size corrections to the dispersion relation and transverse mode profile. Next we applied the method to phonons in an expanding ring shaped condensate. Dimensional reduction with a spatially constant but time dependent transverse profile quickly reproduced previously obtained results [20] for frequency redshifting and "Hubble friction" (amplitude damping). This analysis takes into account the time dependence of the condensate via scaling relations—applicable to power law potentials—for the chemical potential, density, and transverse shape, which we derived for the instantaneous Thomas-Fermi ground state as a function of the radius of the ring potential. We found that the Hubble friction is given by $\dot{\mathcal{V}}/\mathcal{V}$, where $\mathcal{V}$ is the condensate volume, which closely parallels that for a scalar field in an expanding cosmological spacetime.

We next supported the simple analysis for the expanding ring with a systematic transverse mode expansion. Some effects due to time dependence of the background were absorbed by adopting coordinates that are locally comoving with the condensate. Even so, the modes are time-dependent, which significantly complicates the analysis because it allows for mode-mixing transitions. Restricting to the thin-ring limit, we established quantitatively the size of the mode mixing corrections to the simple leading order analysis, find that these transition rates are small compared to the Hubble friction under the conditions of the experiment of [19]. For illustration we applied this method in some detail to the case of a two-dimensional, harmonic thin ring.

In Appendix A, we proved that the velocity in a finite, simply connected condensate is uniquely determined by the density and the atom number conservation equation. In Appendix B, we investigated and characterized the size of deviations from the instantaneous Thomas-Fermi ground state, due to the acceleration on the condensate, for harmonic, box, and vertical harmonic and radial quartic potential that most closely matches the experiment of Ref. [19].

The methods used in this paper are quite generally applicable to condensates in differently shaped or time-dependent trapping potentials. For example, in a periodically modulated transverse potential, the resulting variation of the condensate volume will produce parametric amplification of a resonant longitudinal mode [25] via the $\dot{\mathcal{V}}/\mathcal{V}$ term in (4.21). Depending on experimental interest, it could be valuable to use perturbative methods to extend our time dependent results beyond the thin ring limit studied here. In the other direction, condensates whose transverse size in one or two dimensions is not large compared to the healing length are also of interest. Our results do not apply directly to these, since here we neglected the quantum potential in all considerations. Condensates with such highly confined geometries have been studied using similar methods [23, 26, 27], although not in the context of time dependent trapping potentials, apart from those with a moving step used in black hole and Hawking radiation analog experiments [5–7]. In that setting, a one-dimensional reduction of the GP equation with an effective interaction constant has been exploited in extensive theoretical studies, but as far as we know a systematic analysis of the corrections to that description has not been carried out.

## Acknowledgements

For helpful correspondence we are grateful to J. M. Gomez Llorente and J. Plata, as well as to I. Carusotto, U. Fischer, R. Parentani, and I. Rothstein. We also thank H. Sosa Martinez for helpful comments on the manuscript. This research was supported in part by NSF grant PHY-1708139.

## A Determination of condensate velocity from density

The velocity field $v^i = -(\hbar/M)h^{ij}\partial_j\phi$ for a solution of the GP eqution is uniquely determined by the density, together with the atomic number conservation equation,

$$\partial_t(\sqrt{h}n) + \partial_i(\sqrt{h}nv^i) = 0. \tag{A.1}$$

To prove this, note first that, if two irrotational velocity fields $v_1^i$ and $v_2^i$ have the same circulation around noncontractible loops, then they can only differ by the gradient of some globally defined function $f$. If moreover the density is the same for both vector fields, then (A.1) implies that $\partial_i(\sqrt{h}nh^{ij}\partial_j f) = 0$. Multiplying this equation by $f$, and integrating over the volume of the condensate, we find after integration by parts that $\int \sqrt{h}nh^{ij}(\partial_i f)(\partial_j f) = 0$. Because $n = 0$ at the boundary, the integration by parts produces no boundary term. The integrand is everywhere nonnegative, since $n > 0$ in the condensate and $h^{ij}\partial_i f\,\partial_j f \geq 0$. It follows that the integrand vanishes everywhere, which implies that $\partial_i f = 0$, and thus that $v_1^i = v_2^i$. In particular, if $n\sqrt{h}$ is time independent in some coordinate system, then the unique solution to the continuity equation (A.1) is $v^i = 0$ in that coordinate system.

# B   Dynamics of the condensate due to expansion

In the body of the paper we placed the background condensate in the instantaneous ground state at each time. In this Appendix, we consider dynamical effects on the condensate due to expansion of the ring potential, and how they modify the scalings with ring radius of the of the condensate volume $\mathcal{V}$ and azimuthal sound speed $c_\theta$ that were studied in Sec. 4.

The time dependence of the expanding ring potential of course causes the BEC to acquire time dependence. In the main text we made the simplifying assumption that the BEC at each moment is in the ground state corresponding to the instantaneous applied potential. Here we first make an adiabatic approximation, taking the inertial forces into account by adding them to the applied potential, and estimating how they modify the instantaneous ground state. Under this approximation, we study analytically two simple applied potentials, and then evaluate the scalings numerically for the potential used in the experiment of Ref. [19]. Next we conclude by considering deviations from adiabatic motion of the condensate, due to coherent excitation of the first radial Bogoliubov mode. We find that, to a good approximation, the scaling behavior is accurately captured by the static assumption adopted in the main text.

**Adiabatic motion**

In the adiabatic approximation, the BEC adjusts to the changing applied plus inertial potential, so that it possesses a space and time dependent local flow velocity. The dominant velocity component is clearly the radial expansion $\dot{R}$, but that is not the only velocity. As a result of the dilution of the expanding condensate density, the ring cross section contracts, relative to its expanding center, which means that the BEC velocity has nonzero $\rho$ and $z$ components. The leading contribution from this "contraction velocity" $v$ is suppressed compared to that from $\dot{R}$ by the ratio $v/\dot{R}$. To estimate $v$ we use the scaling of the condensate configuration that would hold if the condensate were at rest at each radius. In Sec. 4.1 we showed that $\rho_\mu$ (for example) scales with $R$ as $\rho_\mu \propto R^{-\alpha/n_\rho}$, from which it follows that $\dot{\rho}_\mu/\rho_\mu \sim (\alpha/n_\rho)\dot{R}/R$, so that $v_\rho \sim \dot{\rho}_\mu \sim (\alpha/n_\rho)(\rho_\mu/R)\dot{R}$. In the experiment of [19], $\alpha/n_\rho \sim 1/7$, and at the smallest ring radius $\rho_\mu/R \lesssim 2/3$, hence $v/\dot{R}$ is no larger than $\sim 0.1$ during the expansion. Thus, for the purposes of the TF approximation, at the precision we are working with, these contraction velocity contributions can be neglected.

To analyze the effects of the bulk radial motion, we begin by writing the action (2.1) using cylindrical coordinates $(r,z,\theta)$, and transforming to the radially comoving coordinate $\rho \equiv r - R(t)$. In cylindrical coordinates, the action takes the form

$$\int dt\,d\theta\,dz\,dr\,r\Big\{i\hbar\Psi^*\partial_t\Psi - V(\rho,z)\Psi^*\Psi - \frac{g}{2}|\Psi|^4$$
$$- \frac{\hbar^2}{2M}[\partial_r\Psi^*\partial_r\Psi + \partial_z\Psi^*\partial_z\Psi + r^{-2}\partial_\theta\Psi^*\partial_\theta\Psi]\Big\}, \tag{B.1}$$

while in comoving coordinates $(\rho,z,\theta)$ it becomes

$$\int dt\,d\theta\,dz\,d\rho\,(R+\rho)\Big\{i\hbar\Psi^*(\partial_t - \dot{R}\partial_\rho)\Psi - V(\rho,z)\Psi^*\Psi - \frac{g}{2}|\Psi|^4$$
$$- \frac{\hbar^2}{2M}[\partial_\rho\Psi^*\partial_\rho\Psi + \partial_z\Psi^*\partial_z\Psi + (R+\rho)^{-2}\partial_\theta\Psi^*\partial_\theta\Psi]\Big\}. \tag{B.2}$$

To eliminate the $\dot{R}\partial_\rho\Psi$ term, we factor a particular local phase out of the wavefunction,

$$\Psi =: e^{iM\eta/\hbar}\bar{\Psi}, \qquad \eta := \dot{R}\rho + \frac{1}{2}\int^t \dot{R}^2 dt'. \tag{B.3}$$

When the action is expressed in terms of $\bar{\Psi}$, the $\dot{R}\partial_\rho\Psi$ term is cancelled, and the result is

$$\int dt\,d\theta\,dz\,d\rho\,(R+\rho)\left\{i\hbar\bar{\Psi}^*\partial_t\bar{\Psi}-\left[V(\rho,z)+m\ddot{R}\rho-i\frac{\hbar}{2}\frac{\dot{R}}{R+\rho}\right]\bar{\Psi}^*\bar{\Psi}-\frac{g}{2}|\bar{\Psi}|^4\right.$$
$$\left.-\frac{\hbar^2}{2M}[\partial_\rho\bar{\Psi}^*\partial_\rho\bar{\Psi}+\partial_z\bar{\Psi}^*\partial_z\bar{\Psi}+(R+\rho)^{-2}\partial_\theta\bar{\Psi}^*\partial_\theta\bar{\Psi}]\right\}.\tag{B.4}$$

Two new terms have been introduced: the "inertial potential" $m\ddot{R}\rho$, and an imaginary contribution to the effective potential, which is $i\hbar/2$ times the fractional rate of change of the measure, $\partial_t\sqrt{h}/\sqrt{h}=[\partial_t(R+\rho)]/(R+\rho)=\dot{R}/(R+\rho)$. While it might seem as if this imaginary term spoils unitarity, since the Hamiltonian is no longer Hermitian, it is actually precisely what is necessary to preserve unitarity when the Hilbert space inner product is time dependent [28]. Of course, in our problem, we could just use the original Cartesian or cylindrical coordinates, and avoid the appearance of time-dependence in the inner product.

The imaginary potential can be absorbed by a time and space dependent rescaling of the wavefunction,

$$\bar{\Psi}=(R+\rho)^{-1/2}\hat{\Psi}.\tag{B.5}$$

The probability density is $(R+\rho)^{-1}\hat{\Psi}^*\hat{\Psi}$, so that the changing measure $R+\rho$ in the inner product is compensated by the prefactor $(R+\rho)^{-1}$. The action in terms of $\hat{\Psi}$ takes the form[10]

$$\int dt\,d\theta\,dz\,d\rho\left\{i\hbar\hat{\Psi}^*\partial_t\hat{\Psi}-\left[V(\rho,z)+m\ddot{R}\rho\right]\hat{\Psi}^*\hat{\Psi}-\frac{g}{2(R+\rho)}|\hat{\Psi}|^4\right.$$
$$\left.-\frac{\hbar^2}{2M}\left[\left|\partial_\rho\hat{\Psi}-\frac{1}{2(R+\rho)}\hat{\Psi}\right|^2+|\partial_z\hat{\Psi}|^2+\frac{1}{(R+\rho)^2}|\partial_\theta\hat{\Psi}|^2\right]\right\}.\tag{B.7}$$

Apart from eliminating the imaginary potential, the rescaling resulted in addition of the term $-\frac{1}{2(R+\rho)}\hat{\Psi}$ to $\partial_\rho\hat{\Psi}$, which is not an important complication since the contribution of this quantity is actually quite small, relative to the other terms in the equations of motion. Specifically, its contributions are at most of relative order $\sim(\xi_c/R)^2$ and $\sim(\xi_c/R)(v_\rho/c_c)$, where $\xi_c=\hbar/2mc_c$ is the healing length at the center of the condensate, and $v_\rho$ is the radial velocity relative to the expanding central radius of the trap potential.

Dropping the kinetic terms in (B.7), the action becomes

$$\int dt\,d\theta\,dz\,d\rho\left\{i\hbar\hat{\Psi}^*\partial_t\hat{\Psi}-\left[V(\rho,z)+M\ddot{R}\rho\right]\hat{\Psi}^*\hat{\Psi}-\frac{g}{2(R+\rho)}|\hat{\Psi}|^4\right\},\tag{B.8}$$

and the variation of $\hat{\Psi}^*$ then yields a time dependent TF equation,

$$i\hbar\partial_t\hat{\Psi}=\left[V(\rho,z)+M\ddot{R}\rho\right]\hat{\Psi}+gn\hat{\Psi},\tag{B.9}$$

where we used that $|\hat{\Psi}|^2/(R+\rho)=|\Psi|^2=n$ is just the atom density. To find the instantaneous ground state wavefunction in the TF approximation, we assume the time dependence of $\hat{\Psi}$ is

---

[10]To avoid the additional terms that arise from the $\partial_\rho$ derivative hitting the $\rho$ in the prefactor $(R+\rho)^{-1/2}$, we could instead factor out only $R^{-1/2}$. This would simplify the kinetic energy term with the $\rho$ derivatives, and would still eliminate the leading order effect of the imaginary term in the thin ring limit, $\rho/R\ll1$. In terms of $\check{\Psi}:=R^{1/2}\bar{\Psi}$ the action takes the form

$$\int dt\,d\theta\,dz\,d\rho\,(1+\rho/R)\left\{i\hbar\check{\Psi}^*\partial_t\check{\Psi}-\left[V(\rho,z)+M\ddot{R}\rho+i\frac{\hbar}{2}\frac{\dot{R}}{R}\frac{\rho}{R+\rho}\right]\check{\Psi}^*\check{\Psi}-\frac{g}{2R}|\check{\Psi}|^4\right.$$
$$\left.-\frac{\hbar^2}{2M}[\partial_\rho\check{\Psi}^*\partial_\rho\check{\Psi}+\partial_z\check{\Psi}^*\partial_z\check{\Psi}+(R+\rho)^{-2}\partial_\theta\check{\Psi}^*\partial_\theta\check{\Psi}]\right\}.\tag{B.6}$$

trivial: $i\hbar\partial_t\hat{\Psi} = \hat{\mu}\hat{\Psi}$, for some spatially constant but time dependent "chemical potential" $\hat{\mu}$. Then (B.9) becomes

$$\hat{\mu} = V(\rho,z) + M\ddot{R}\rho + gn, \tag{B.10}$$

which is the usual TF equation for the ground state density, including the inertial contribution to the potential. Note that no thin ring approximation was made in obtaining this equation.

To evaluate the impact of the inertial potential $M\ddot{R}$, let us first consider a harmonic trapping potential of the form $V(\rho,z) = M\omega_\rho^2\rho^2/2 + M\omega_z^2 z^2/2$. The inertial potential $M\ddot{R}\rho$ shifts the minimum of the effective potential $V_{\text{eff}} := V(\rho,z) + M\ddot{R}\rho$ relative to that of $V(\rho,z)$:

$$V_{\text{eff}}(\rho,z) = M\omega_\rho^2(\rho + a_\rho)^2/2 + M\omega_z^2 z^2/2 - M\omega_\rho^2 a_\rho^2/2, \tag{B.11}$$

where $a_\rho = \ddot{R}/\omega_\rho^2$.[11] In terms of the shifted chemical potential $\tilde{\mu} := \hat{\mu} + M\omega_\rho^2 a_\rho^2/2$, the Thomas-Fermi condensate density takes form, $gn = \tilde{\mu} - V(\rho + a_\rho, z)$, and the integral for the total number of atoms $N$ becomes

$$gN = 2\pi\tilde{\mu}\int d\rho\, dz\,(R+\rho)\left[1 - \left(\frac{\rho+a_\rho}{\rho_{\tilde{\mu}}}\right)^2 - \left(\frac{z}{z_{\tilde{\mu}}}\right)^2\right], \tag{B.12}$$

where $\tilde{\mu}/\rho_{\tilde{\mu}}^2 := M\omega_\rho^2/2$, and similarly for $z_{\tilde{\mu}}$. With the change of variables to $\bar{z} = z/z_{\tilde{\mu}}$ and $\bar{\rho} = (\rho + a_\rho)/\rho_{\tilde{\mu}}$, and defining the shifted radius of the condensate by $R_c := R - a_\rho$, this integral becomes

$$gN = 2\pi R_c\rho_{\tilde{\mu}}z_{\tilde{\mu}}\tilde{\mu}\int d\bar{\rho}\,d\bar{z}\,(1+\tilde{\zeta}\bar{\rho})\left[1 - \bar{\rho}^2 - \bar{z}^2\right], \tag{B.13}$$

where $\tilde{\zeta} := \rho_{\tilde{\mu}}/R_c$. This is identical to the number integral (4.8), after the replacements $R \to R_c$ and $\mu \to \tilde{\mu}$. Thus, in a harmonic potential, we can absorb completely the effect of the inertial potential by scaling the various quantities by powers of the radius of the moving effective potential $R_c$ rather than of $R$.

Let us also consider the other extreme: the box potential for a two dimensional condensate ($n_\rho \to \infty$), with full width of the box $2\rho_w$. Assuming that the acceleration is small enough that $M\ddot{R}\rho_w < \hat{\mu}$, then the ground state of the applied plus inertia potential extends from $\rho = -\rho_w$ to $\rho = \rho_w$. In this case the total area and cross sectional length of the 2D condensate are unaffected by the acceleration, and the effective azimuthal sound speed (4.19) is given by

$$c_\theta^2 = \frac{1}{2\rho_w M}\int_{-\rho_w}^{\rho_w} d\rho\,\frac{\hat{\mu} - M\ddot{R}\rho}{1 + \rho/R}. \tag{B.14}$$

In terms of the parameter $\alpha_\mu := M\ddot{R}\rho_w/\hat{\mu}$, quantifying the ratio of the inertial potential to the chemical potential, and the ring thickness parameter $\zeta = \rho_w/R$, (B.14) yields

$$c_\theta^2 = \frac{\hat{\mu}}{M}[1 + (\alpha_\mu\zeta + \zeta^2)/3 + O(\alpha_\mu\zeta^3, \zeta^4)]. \tag{B.15}$$

As for the chemical potential $\hat{\mu}$, the integral for the total number of atoms $N$ becomes

$$gN = 2\pi\int_{-\rho_w}^{\rho_w}(R+\rho)\left(\hat{\mu} - M\ddot{R}\rho\right)d\rho, \tag{B.16}$$

which yields

$$\hat{\mu} = \mu_0\frac{1}{1 - a_\mu\zeta/3}, \tag{B.17}$$

---

[11] For the experiment of Ref. [19], $a_\rho \approx 3.2\ \mu$m for the $R_f/R_i = 1.9$ expansion and $a_\rho \approx 2.6\ \mu$m for the $R_f/R_i = 4.1$ expansion.

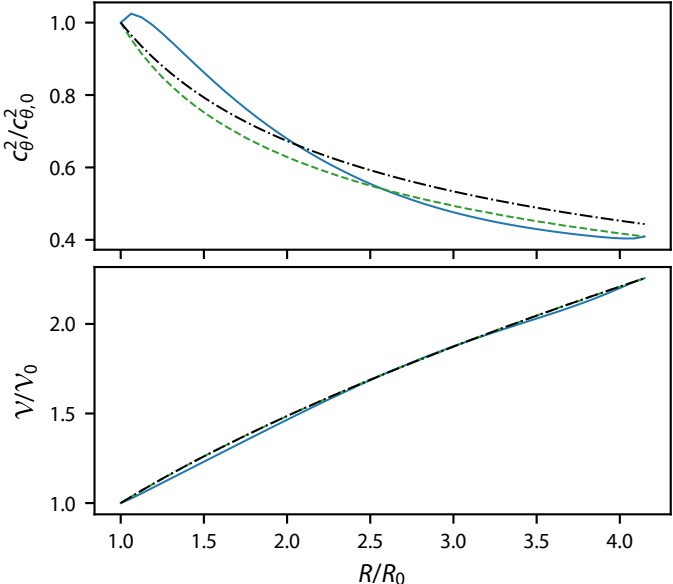

Figure 1: Scalings of azimuthal sound speed squared (top) and volume (bottom) for the instantaneous ground state of the condensate in Ref. [19]. The dot-dashed curves are for the thin ring limit, without the inertial potential ($c_\theta^2 \propto R^{-4/7}$ and $\mathcal{V} \propto R^{4/7}$). The dashed curves include the ring thickness, while the solid curves include both ring thickness and the inertial potential.

where $\mu_0 := gN/4\pi R\rho_w$ is the chemical potential for the stationary ring at the same radius. Through second order in the parameters $\alpha_{\mu_0}$ and $\zeta$, the squared azimuthal sound speed (B.15) is thus given by

$$c_\theta^2 \approx \frac{\mu_0}{M}[1 + (2\alpha_{\mu_0}\zeta + \zeta^2)/3]. \tag{B.18}$$

The effect of the acceleration on $c_\theta^2$ is thus suppressed by the ring thickness parameter. The product $\alpha_{\mu_0}\zeta$ is largest at the start of the ring's expansion, since $\zeta$ is largest then and $\alpha_{\mu_0}$ is roughly antisymmetric in time about the midpoint. For the experiment of Ref. [19], the maximum $\alpha_{\mu_0}\zeta \approx 0.2$,[12] and thus its effect on $c_\theta^2$ is $\lesssim 13$ %. The effect of the $\zeta^2$ term is already included in (4.19), and here we see that its effect is also $\lesssim 13$ %.

To more accurately quantify the adiabatic noninertial and ring thickness effects for the experiment of Ref. [19], we have numerically evaluated the scalings of $\mathcal{V}$ and $c_\theta^2$, for the instantaneous ground state with the quartic radial potential and quadratic vertical potential that most closely matches the experiment of Ref. [19]. Fig. 1 shows plots of these quantities, relative to their initial values (denoted by the zero subscript), vs. $R/R_0$. The dot-dashed curves are for the simplest approximation, the thin ring limit and without the inertial potential. The dashed curves include the ring thickness effect, while the solid curves include both ring thickness and the inertial potential. The acceleration affects the volume by at most 3 %, while it affects $c_\theta^2$ by at most about 15 %.

---

[12]For the experiment of Ref. [19] and using $\rho_w = \rho_\mu$, $\alpha_{\mu_0} \approx 0.45 \to 0.6$ for the $R_f/R_i = 1.9$ expansion and $\alpha_{\mu_0} \approx 0.25 \to 0.45$ for $R_f/R_i = 4.1$ expansion, and the values of $\zeta$ are given in footnote 6. The largest value of $a_{\mu_0}\zeta$ occurs at the beginning of the $R_f/R_i = 1.9$ expansion.

**Non-adiabatic evolution**

Before concluding that (B.10) forms a suitable foundation for estimating the evolving shape of the condensate, there is one more effect that should be considered. Namely, the condensate responds dynamically to the variable acceleration of the ring potential, and is generally offset from what would be the instantaneous ground state at each radius, and time dependent. The most dramatic effect in this regard happens at the end of the expansion of the potential, when the condensate "overshoots" and sloshes back, forming one or more dark solitons, which subsequently decay into a turbulent state. Prior to that, however, the effect is less dramatic. The condensate first lags behind the ring potential, and later runs ahead of it. This behavior can be seen in Fig. 3 of Ref. [19].

This dynamics of the condensate corresponds to radial Bogoliubov mode excitations added to the adiabatic background motion described above. The most excited will be the lowest mode, which, in the thin ring limit, yields $\delta n \sim \bar{\rho}$ for the harmonic potential, and $\delta n \propto \sin(\pi \bar{\rho} /2)$ for the box potential. In both cases, the mode is antisymmetric under $\bar{\rho} \to -\bar{\rho}$, so to first order in the amplitude it produces no change in the instantaneous value of $c_\theta^2$. Any effect would thus be higher order in the excited mode amplitude, and/or would be suppressed by the ring thickness parameter. Moreover, we note that in addition any effect that is linear in the mode amplitude will average to zero over a period of oscillation, if the ring potential radius does not change during the oscillation.

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
