# Peer review of "Phonon redshift and Hubble friction in an expanding BEC"

_SciPost Physics, doi:SciPost Phys. 10, 064 (2021)_

## Round 2 · Referee Report · Anonymous (Referee 1) · 2020-10-9

Strengths

The paper is clearly written in a rather pedagogical way that allows even not expert in the field to appreciate it.

Report

The authors discuss in a detailed way the phonon wave equation in an expanding
ring BEC by dimensional reduction , showing characteristic effects one finds in cosmology , namely redshifting and Hubble friction experienced by quantum fields in an expanding universe.

---

## Round 2 · Referee Report · Anonymous (Referee 2) · 2020-10-23

Strengths

A perturbative analysis is introduced to analyse azimuthal phonons' dynamics in an expanding ring BEC. This analysis is first applied to the case of a static cylindrical condensate (where known results are recovered), and can also be useful to address scenarios more complicated than those considered.

Weaknesses

The analysis is rigorous and thorough, I do not see weaknesses.

Report

This is a technically very valuable paper. In the context of the gravitational analogy the authors first write down a 3D action for the linearised perturbations (phonons) in a BEC. Then, for both static cylindrical and expanding ring condensates, they derive dimensionally reduced 1D actions for phonon fields phi_1 constant in the transverse dimensions. In the latter case, the phonon field wave equation is similar to that of a scalar field in an expanding universe.
Their perturbative analysis allows to compute the corrections due to transverse modes, and the results show that these corrections are small. In the case of the expanding ring BEC, they improve their theoretical predictions with respect to their experiment [19], also in connection with the published results in [20].
In my opinion, after the authors have considered the two (small) points below, both for the analysis and the results the paper meets publishing criteria.

Requested changes

A couple of points: - In section 4.1, for the expanding ring BEC, they mention that the static approximation is surprisingly accurate even though in [19] the condition \dot R << c is violated (and they show in the appendix that the corrections to the static approximation are indeed small). Do the authors understand why ? - in the definition of the speed of sound c_theta^2 in (4.19), isn't the factor (1+rho/R) in the numerator?

---

## Round 3 · Referee Report · Anonymous (Referee 1) · 2020-11-15

Strengths

Clear paper

Weaknesses

None

Report

The authors discuss in a detailed way the phonon wave equation in an expanding ring BEC by dimensional reduction , showing characteristic effects one finds in cosmology , namely redshifting and Hubble friction experienced by quantum fields in an expanding universe. So far the study of analogue gravity models were mostly devoted to find the analogue of Hawking radiation in black holes. The application to cosmological issues is a developing field extremely interesting. The paper is clearly written in a rather pedagogical way that allows even not expert in the field to appreciate it.

---

## Round 3 · Referee Report · Anonymous (Referee 2) · 2020-11-16

Strengths

Clear and interesting

Weaknesses

No weaknesses

Report

The authors successfully addressed the points I raised in my report, I recommend publication of their manuscript.

---

## Round 3 · Referee Report · Matthew Davis (Referee 3) · 2021-2-13

Strengths

  1. Presents an action-based method that helps understand the results of an important experiment on analogue cosmology.
  2. Paper is comprehensive, and clearly set out the details of the calculations.
  3. Informative abstract/introduction/conclusion.

Weaknesses

  1. Perhaps the manuscript is a little long-winded in the main text, but this is a matter of personal taste.

Report

This manuscript corrects an analysis of the experiment performed in Ref. [19], and hence derives a result for the "Hubble friction" that is in agreement with the experimental results and an alternative theoretical approach of Ref. [20]. It is generally straightforward to follow, and well written. The methodology and dimensional reduction techniques could prove useful in application to alternative scenarios, and will be of interest to several in the field. I don't think there are any further significant changes following those from the first round of refereeing. Therefore I feel that this manuscript meets the acceptance criteria for SciPost Physics.

Requested changes

  1. After Eq. (3.17) there appears $kr_\mu<1$ - I suspect this is the condition for the validity of Eq. ( 3.17)?
  2. The conclusion refers to the "short" and "long" appendix - I think it would be best to say "Appendix A" and "Appendix B"

  • validity: high
  • significance: good
  • originality: good
  • clarity: good
  • formatting: excellent
  • grammar: excellent

Author:  Stephen Eckel  on 2021-02-22  [id 1259]

(in reply to Report 3 by Matthew Davis on 2021-02-13)
Category:
answer to question

We thank Prof. Davis for his helpful comments. In the published version, we will be sure to eliminate the spurious reference to $k r_\mu < 1$ after (3.17) and will properly reference the appendices.

---

## Round 3 · Author Response

Dear Editor,

Thank you for considering our manuscript for publication on SciPost.

We thank the reviewers and include replies to their comments here. Specifically, with respect to the second reviewer:

  • In section 4.1, for the expanding ring BEC, they mention that the static approximation is surprisingly accurate even though in [19] the condition \dot R << c is violated (and they show in the appendix that the corrections to the static approximation are indeed small). Do the authors understand why?

The condition \dot R << c would certainly imply adiabaticity, but it is not strictly required. As discussed in the appendix, the accuracy actually comes from a combination of factors. First, in the adiabatic limit where no additional Bogoliubov modes are excited by the expansion, the effect of the additional non-inertial forces on the relevant azimuthal modes enters at lowest order with the thin ring parameter. Thus, the thin ring used in Ref. [19] helps to suppress any effect. For some of the expansions in [19], there is clear evidence of non-adiabatic evolution in the form of oscillations of the center of mass of the condensate after the expansion stops. The effect of this non-adiabatic motion is suppressed by of the odd symmetry of the first radial Bogoliubov mode, which has no impact on the relevant azimuthal phonon modes at first order in the excitation amplitude.

In our resubmission, we have updated the statement regarding the condition \dot R << c to make it clear that while it is sufficient for adiabaticity, it is not strictly required. We also summarized the arguments of the appendix in a new, expanded discussion at the start of Sec. 4.2.

  • in the definition of the speed of sound c_theta^2 in (4.19), isn't the factor (1+rho/R) in the numerator?

Referring to Eq. (2.5), there are two contributions to this factor: r from the measure of the integrand \sqrt{-h} and 1/r^2 from the inverse metric h^{ij}. Combined, these two produce a factor of 1/r = 1/(1+\rho/R).

In our resubmission, we have updated the definitions after (2.1) in order to make explicit the distinction between the covariant and contravariant (inverse) components of the metric tensor.

With these changes, we respectfully resubmit our improved manuscript for your consideration.

Best regards,

Stephen Eckel and Ted Jacobson

---

## Round 3 · List of Changes

As a result of the reviewer comments, we have made the following changes to the manuscript: 1. We have updated the definitions after (2.1) in order to make explicit the distinction between the covariant and contravariant (inverse) components of the metric tensor. 2. We have updated the statement regarding the condition \dot R << c to make it clear that while it is sufficient for adiabaticity, it is not strictly required. We also summarized the arguments of the appendix in a new, expanded discussion at the start of Sec. 4.2.

In addition to these two changes, we also updated the text after (B.11) to remove a superfluous double minus sign and added a footnote with the calculated $a_\mu$ parameters of Ref. [19]. These two changes help to make the argument of the appendix clearer for the reader.

---

## Editorial Decision

published